# *Colletotrichum truncatum*—A New Etiological Anthracnose Agent of Sword Bean (*Canavalia gladiata*) in Southwestern China

**DOI:** 10.3390/pathogens11121463

**Published:** 2022-12-02

**Authors:** Min Shi, Shi-Ming Xue, Mei-Yan Zhang, Shi-Ping Li, Bi-Zhi Huang, Qi Huang, Qiong-Bo Liu, Xiang-Long Liao, Yan-Zhong Li

**Affiliations:** 1State Key Laboratory of Herbage Improvement and Grassland Agro-Ecosystems, Ministry of Agriculture and Rural Affairs, Engineering Research Center of Grassland Industry, Ministry of Education, College of Pastoral Agriculture Science and Technology, Lanzhou University, Lanzhou 730000, China; 2Academe of Grassland and Animal Science, Kunming 650212, China; 3Forage and Fodder Station of Qujing, Qujing 655000, China

**Keywords:** pathogen, *Colletotrichum truncatum*, sword bean, pathogenicity, morphology, phylogenetic analysis

## Abstract

Anthracnose is a disease caused by *Colletotrichum* species. They are well known as major plant pathogens, and a black stem disease, specifically caused by *Colletotrichum truncatum* and primarily infecting sword bean (*Canavalia gladiata*), was observed in the Yunnan province, China. To aid disease management and to determine pathogenic characteristics, the species causing the leaf spot disease of hairy vetch was verified as *C. truncatum*. A sequence analysis of the ITS, ACT, GAPDH, and HIS3 genes was conducted, as well as morphological and cultural characteristics, to identify this *Colletotrichum* species, which has curved conidia. *C. truncatum* isolates from sword bean formed a distinctive group among *Colletotrichum* species, including those that infect other forage and field crops. Artificially inoculated sword bean seedlings showed typical symptoms of anthracnose, which were similar to field observations. To the best of our knowledge, this is the first report of *C. truncatum* causing black stem disease on sword beans in China.

## 1. Introduction

Sword bean (*Canavalia gladiata*) is an important annual legume consumed as a vegetable and as a medicine, and is endemic to tropical Asia, Africa, and India [1,2]. It is currently cultivated across 0.065 million hectares in southwestern China [3,4]. All parts of the sword bean are edible, such as the beans, leaves, pods, or roots [5]. Immature sword bean pods are extensively utilized in Asia as a vegetable [6]. It is widely cultivated across the world, but especially so in central and southern China, where it is an important food supply [4]. It can increase soil nutrient levels, improve soil structure, and reduce soil erosion [7]. Sword bean is the third most important medicinal and food crop in southwestern China.

The fungal pathogen *Colletotrichum* causes anthracnose on legume crops globally [8,9,10]. The genus *Colletotrichum* includes about 600 species that can destroy many crops [11], and cause typical symptoms such as sunken necrotic lesions, generally known as anthracnose [12]. Anthracnose is an important disease of sword bean, and *Colletotrichum capsica* and *Colletotrichum lindemuthianum* have specifically been detected in sword beans in India and China [13,14].

*Colletotrichum truncatum* is a species commonly reported to cause disease in papaya, lentils, and soybeans [15,16,17,18]. No previous studies have reported any fungi associated with anthracnose disease on sword beans. Therefore, this study aimed to identify the causal agent of anthracnose disease in sword beans by investigating its morphological and molecular properties, as well as to test its pathogenicity.

## 2. Materials and Methods

### 2.1. Plant Collection and Fungal Isolation

A total of fifteen diseased stems from five plants (three stems per plant) were excised from the margins of lesions. Stem pieces were sterilized with 75% ethanol for 30 s and 1% NaClO for 75 s, rinsed three times with sterile distilled water, dried three times between sterile filter paper, and plated onto potato dextrose agar (PDA) supplemented with 25 mg/L of penicillin and streptomycin, and incubated at 24 °C for 3 to 21 days [19]. The isolation frequency for each tissue type was determined by examining colony characteristics on the fourth day. Pure colonies were obtained by transferring 1–2 mm emerging hyphal tips onto fresh PDA plates with no supplements. A diseased specimen, as well as three isolates, was deposited at the Mycological Herbarium of the Lanzhou University (MHLZU) with the numbers MHLZU19328, and YN1932501, YN1932502, YN1932503, and YN1932504, respectively.

### 2.2. Morphological Characterization

Mycelial plugs (5 mm in diameter) were removed from subcultured hyphae colony edges and dark-incubated at 25 °C on PDA. The PDA colony characteristics were recorded after 10 days, and acervuli, setae, and conidia were observed using a stereomicroscope (Nikon ECLIPSE Ti, Tokyo, Japan) and regular microscope and photographed with a Canon DS126391 camera (Canon, Lanzhou, China). Colony diameters and colors were recorded from PDA-grown cultures.

### 2.3. DNA Extraction, Polymerase Chain Reaction (PCR) Amplification, and Sequencing

Pure culture mycelia were scraped into centrifuge tubes using a sterilized spoon. Total genomic DNA was extracted from representative isolates using a Fungal DNA Kit (D3195; OMEGA Biotech Co. Ltd., Norcross, GA, USA) following the manufacturer’s instructions. DNA samples were stored at −20 °C for further study. The rDNA internal transcribed spacer (ITS), a partial actin sequence (ACT), glyceraldehyde-3-phosphate dehydrogenase (GAPDH), and histone3 (HIS3) genes were amplified (PCR amplification and sequencing primers are given in Table 1). PCR reactions were performed in a 2720 Thermal Cycler (Applied Biosystems, Foster City, CA, USA) in a total volume of 25μLthat contained 1 μL of genomic DNA, 1 μL of forward and reverse primers, 12.5 μL of 2 x High-Fidelity Master Mix, and 9.5 μL of ddH_2_O. The PCR conditions were as follows: an initial denaturation step was performed at 94 °C for 3 min, followed by 30 cycles at 94 °C for 10 s, then 30 s of annealing (54 °C, 56 °C, 46 °C, 56 °C, and 52 °C for ITS, ACT, GAPDH, and HIS3, respectively), and 72 °C for 10 s. A final extension step was performed at 72 °C for 10 min [20].

### 2.4. Phylogenetic Analysis

The sequences obtained from the four isolates, as well as other *Colletotrichum* spp. reference sequences as described by Damm [26], were downloaded from GenBank (Table 2).

Single sequences were aligned using ClustalW in MEGA 5.1. The four loci were combined with Sequence Matrix 1.8. The best-fit nucleotide substitution models of each gene were assessed by MrModeltest 2.3 for Bayesian reference (BI) analysis. The best-fit model of each gene was imported into MrBayes v. 3.2.6, and the full dataset was run for 2,000,000 generations, and sampled every 100 generations and four chains. The resulting tree was created in Figtree v1.4.3, Adobe Acrobat DC (Adobe, San Jose, CA, USA), and Microsoft Office PowerPoint 2007 (Microsoft, Redmond, CA, USA).

### 2.5. Pathogenicity Assay

The pathogenicity of *C. truncatum* was determined by a spray treatment. A total of 100 sword bean seeds were obtained from sword bean plants during the 2020 harvest season, surface-sterilized with 75% ethanol for 30 s and 1% NaClO for 75 s, rinsed three times with sterile distilled water, and transferred into sterilized Petri dishes that contained a double filter paper layer. A total of 5 mL sterile distilled water was added to the seeds, whereafter they were incubated in the dark at 25 °C for 7 days. The fungal strains were cultivated in the dark at 25 °C for 14 days, and the colony surfaces were gently scraped with a glass spreader. The conidial suspension was adjusted to a concentration of 1.0 × 10^6^ conidia/mL sterile water using a hemocytometer. Fifty sword bean plants were transferred into 10 pots (with five plants per pot) that contained 500 g of sterilized soil. The stems of 50 healthy plants were sprayed with a conidial suspension that contained 1.0 × 10^6^ conidia/mL and 0.01% Tween 80. A total of 50 healthy plants were used as controls and received a sterile water spray. All plants were kept in a greenhouse (days: 22 °C, 18 h light; nights: 18 °C, 6 h dark) and were covered with clear polyethylene bags for three days to maintain a high humidity level.

## 3. Results

### 3.1. Symptoms and Fungal Isolation

Severe anthracnose disease symptoms were observed on sword bean plants from 2020 to 2021 between November and December in Yuanmou County (N 25°84′92″, E 101°83′37″), Yunnan Province. Approximately 60% of the stems were infected. Leaf lesions initially appeared on older leaf blade edges, and presented as the round, or nearly round, black spots marked with a red arrow (Figure 1b). Lesions eventually covered entire blades, and brown necrotic spots with a darker border developed on petioles, which had slightly sunken centers. The initial symptoms were characterized by small, chlorotic spots that appeared on the lower stems; stem lesions were spindle-to-fusiform shaped, and brown to dark brown in the middle of the lesions marked with a red arrow (Figure 1a). Finally, the affected stems eventually turned brown, and sunken, brown necrotic spots covered entire stems. Setae were produced from stem pieces on PDA when the humidity reached 100% (Figure 1c). Three isolates were obtained from symptomatic samples, and the average separation rate was 36.27%.

### 3.2. Morphological Characterization

The PDA plate adaxial fungal colony surfaces were whitish-brown at 7 days and 55–63 mm in diameter, whereafter they became gray to pale gray with sparse white aerial surfaces, whereas the reverse sides were an ash-black color. The PDA colonies subsequently turned black with a flocculous mycelium (Figure 2a,b). Acervuli were produced on the PDA at 13 days and were 124–165 mm in diameter (Figure 2c). Setae were linear and growing on spherical acervuli, were dark brown to black, rigid and straight at their bases, and with rounded tips being 52~54 μm long × 1.8~2.1 μm wide (*n* = 50). Conidia were crescent-shaped, hyaline, smooth-walled, aseptate, and slightly curved with parallel walls, and were 1.9~2.7 × 10.8~15.9 μm in size (Figure 2d).

### 3.3. Phylogenetic Analysis

The ITS, ACT, GAPDH, and HIS3 gene sequences were obtained from GenBank. Four representative strains were aligned in MEGA 5.1 for phylogenetic analysis. *C. lindemuthianum* (CBS 315.28) was used as an outgroup. The sequence contained 1449 characters after sequence alignment (429 for ITS, 358 for *ACT*, 270 for *GAPDH*, 392 for *HIS3*). The following models were selected by MrModeltest 2.3 for the MrBayes analysis: GTR+G for ITS, GTR+G for ACT, GTR+I+G for GAPDH, and GTR+G for *HIS3*. The combined multigene tree (Figure 3), including ITS, *ACT*, *GAPDH,* and *HIS3*, showed that the YN1932501, YN1932502, YN1932503, and YN1932504 isolates from this study formed a clade with the representative strains of *C. truncatum* (CBS119189, CBS151.35, and CBS710.70). The corresponding Bayesian posterior probability was 1.0, and the clade was clearly distinct from the other *Colletotrichum* species (Figure 3).

### 3.4. Pathogenicity

To confirm Koch’s postulates, four representative isolates were tested on sword bean stems. Inoculations with YN1932501, YN1932502, YN1932503, and YN1932504 (identified as *C. truncatum*) exhibited spindle to fusiform lesions marked with a red arrow (Figure 4), whereas no symptoms appeared on the control plants. *C. truncatum* was re-isolated from the inoculated plants and identified by morphology as described above.

## 4. Discussion

In this study, sword bean was described as a new host for *C. truncatum* in the Yunnan Province, China. Sunken necrotic lesions were typical symptoms observed on sword bean stems in the field. Koch’s postulates were satisfied since this fungus was (1) isolated from diseased sword bean plants, (2) the successfully inoculated sword bean plants produced the same symptoms, and (3) the fungus was subsequently re-isolated. A combined analysis of the morphological characteristics and multiple gene sequence data further confirmed that all four *Colletotrichum* isolates were *C. truncatum.*

Sword bean anthracnose symptoms caused by *C. truncatum*, as observed in the field, were similar to those caused by *C. lentis*. The stem lesions were initially dark and linear, but later enlarged and often coalesced with adjacent lesions until large stem areas were involved [20]. However, *C. truncatum* anthracnose seemed to cause more and more severe lesions on stems than on leaf laminas, whereas *C. lentis* caused severe lesions on both stems and leaves. Four other plant species grew in the vicinity, namely *Canavalia gladiata, Brachiaria eruciformis, Indigofera amblyantha*, and *Stylosanthes guianensis*, and previous studies have reported that *C. truncatum* can infect *Indigofera* plants. It is possible that the infection source for *C. truncatum* on sword bean might be closely related to the presence of *Indigofera amblyantha* plants, but this must be verified. Furthermore, the climatic conditions of the area, as well as field planting patterns, may also play a role.

Identification based on morphology is a primary step towards classifying fungal pathogens at the genus level [30]. Corda described the genus *Colletotrichum* in 1831, which was historically based on morphological characteristics such as fusiformed, curved, hyaline conidia with acute ends, and brown, opaque, subulate setae with acute tips [31]. The morphology of *C. truncatum* found in this study was similar to that of previous reports [26]. However, *Colletotrichum* species identification based only on morphology is not highly accurate since few morphological characters can distinguish between the species, and the teleomorphic stages are rarely formed [32]. Moreover, morphological species characteristics can change when environmental conditions change, particularly the small morphological differences among *C. destructivum*, *C. linicola*, and *C. truncatum* [33]. The combination of molecular diagnostic tools, along with morphological techniques, is therefore the best approach for studying *Colletotrichum* species complexes [34].

Doyle and Gaut [35] described that single genes are usually insufficient to resolve interspecific relationships in *Colletotrichum*, which is why phylogram trees derived from single genes always yield limited information at lower taxonomic ranks. Using single-gene ITS cannot clearly distinguish between *C. lentis* and *C. truncatum* [29]. Previous studies have indicated that using a combined dataset analysis of datasets to generate multigene phylogenetic trees could provide a higher overall support than any of the single-locus phylogenies [36,37]. In this study, phylogenetic analyses based on the combination of ITS, *ACT*, *HIS3*, and *GAPDH* sequences clearly distinguished *C. truncatum* (YN1932501, YN1932502, YN1932503, and YN1932504) from the other closely related *Colletotrichum* species.

*C. truncatum* has a broad host range and can reportedly infect numerous plant species, such as Chinese flowering cabbage, soybean, lentil, common vetch, alfalfa, chili, solanaceous crops, and hemp plants [17,29,38,39,40]. To the best of our knowledge, no previous record exists of *C. truncatum* causing disease in sword bean. This is, therefore, the first report of sword bean anthracnose caused by *C. truncatum*.

## 5. Conclusions

Anthracnose disease in sword bean plants, caused by *C. truncatum,* was first described in China. The fungi was isolated from infected plants and identified based on the morphological characteristics and molecular properties of multiple DNA sequences. A pathogenicity test revealed similar symptoms in a greenhouse. This is the first report of *C. truncatum* causing anthracnose in sword bean in China. The accurate identification of *C. truncatum* is important for developing efficient control strategies to better understand the epidemiology of this disease. In fact, breeding for resistance against anthracnose depends on it. Further disease management studies are needed to select efficient fungicides for controlling sword bean anthracnose.

## Figures and Tables

**Figure 1 pathogens-11-01463-f001:**
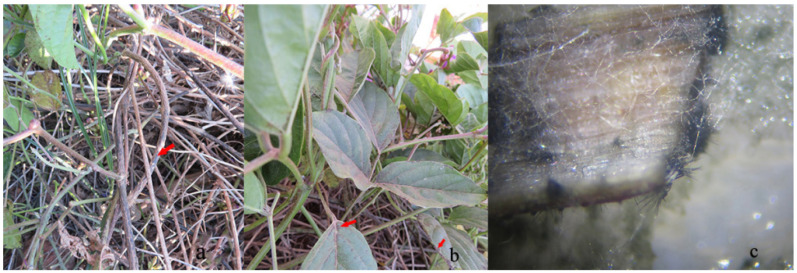
Symptoms of anthracnose caused by curved conidial species of *Colletotrichum* on stem and leaves (**a**) field symptoms of stem spot; (**b**) leaf symptoms of blackening; (**c**) states on stems pieces.

**Figure 2 pathogens-11-01463-f002:**
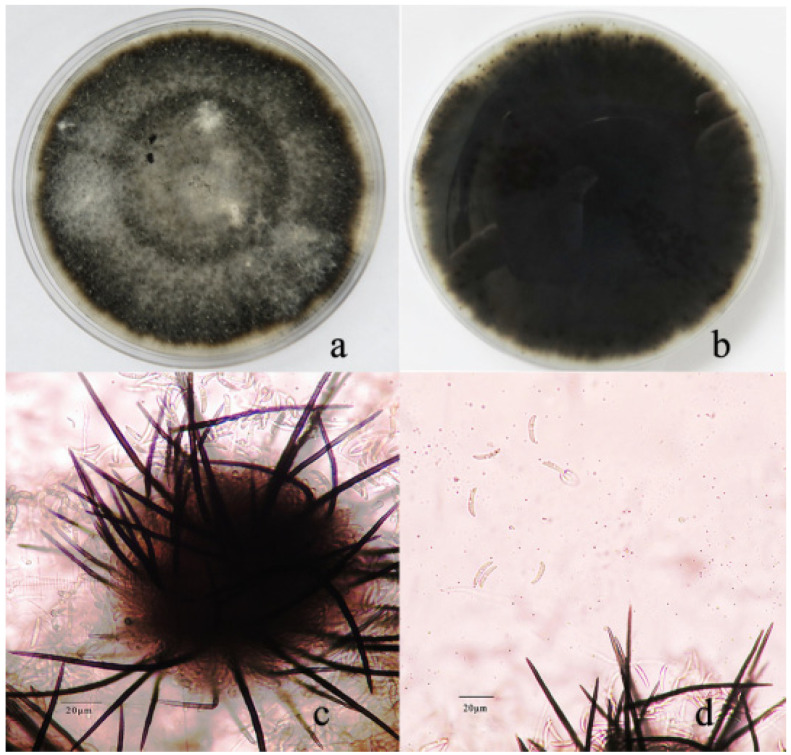
Morphological features of *Colletotrichum truncatum* (YN19325). Colony morphology on PDA ((**a**,**b**); upper and reverse colony). (**c**); acervuli, (**d**) conidia with seta; Scale bar of d = 20 μm.

**Figure 3 pathogens-11-01463-f003:**
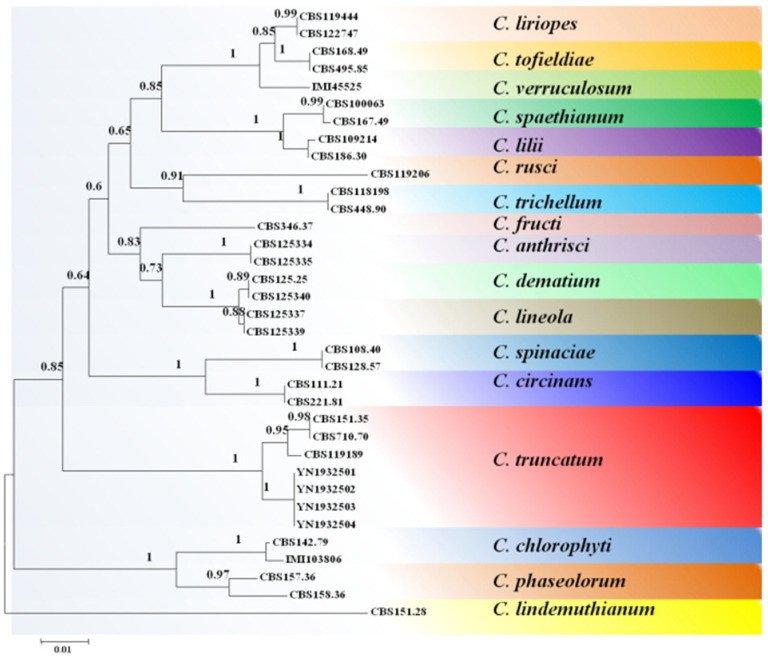
A Bayesian analysis tree of the concatenated partial sequences of ITS, *ACT*, *GAPDH*, and *HIS3*, gene regions of the isolates used in this study. The numbers on the nodes are posterior probability values. Bootstrap support values (100 replicates) above 50% are shown at the nodes. *C. lindemuthianum* CBS 151.28 is used as outgroup.

**Figure 4 pathogens-11-01463-f004:**
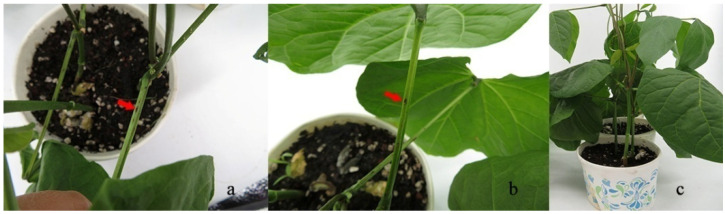
Sword bean stems inoculated with *C. truncatum* after 14 days. (**a**,**b**); sword bean stems inoculated with conidia of *C. truncatum*, (**c**); Control plant.

**Table 1 pathogens-11-01463-t001:** Primers used in this study for PCR and sequencing.

Gene	Product	Primer	Direction	Sequence (5′–3′)	Reference
ITS	Internal transcribed spacer	ITS1	Forward	TCCGTAGGTGAACCTGCGG	[21,22]
		ITS4	Reverse	TCCTCCGCTTATTGATATGC	
*ACT*	Actin	ACT-512F	Forward	ATGTGCAAGGCCGGTTTCGC	[23]
		ACT-783R	Reverse	TACGAGTCCTTCTGGCCCAT	
*GAPDH*	Glyceraldehyde- 3-phosphate dehydrogenase	GDF1	Forward	GCCGTCAACGACCCCTTCATTG	[24]
		GDR1	Reverse	GGGTGGAGTCGTACTTGAGCAT	
*HIS3*	Chitin synthase I	CYLH3F	Forward	AGGTCCACTGGTGGCAAG	[25]
		CYLH3R	Reverse	AGCTGGATGTCCTTGGACTG	

**Table 2 pathogens-11-01463-t002:** Collection details and GenBank accession numbers of isolates.

Species	Cultural Number	Host	Country	GenBank Accessions				
				ITS	ACT	*gapdh*	*His3*	Reference
*C. anthrisci*	CBS 125334	*Anthriscus sylvestris*	The Netherlands	GU227845	GU227943	GU228237	GU228041	[27]
	CBS 125335	*Anthriscus sylvestris*	The Netherlands	GU227846	GU227944	GU228238	GU228042	[27]
*C. chlorophyti*	IMI 103806	*Chlorophytum*	India	GU227894	GU227992	GU228286	GU228090	[28]
	CBS 142.79	*Stylosanthes hamata*	Australia	GU227895	GU227993	GU228287	GU228091	[27]
*C. circinans*	CBS 111.21	*Allium cepa*	USA	GU227854	GU227952	GU228246	GU228050	[26]
	CBS 221.81	*Allium cepa*	Serbia	GU227855	GU227953	GU228247	GU228051	[27]
*C. dematium*	CBS 125.25	*Eryngium campestre*	France	GU227819	GU227917	GU228211	GU228015	[26]
	CBS 125340	*Apiaceae*	Czech	GU227820	GU227918	GU228212	GU228016	[27]
*C. fructi*	CBS 346.37	*Malus sylvestris*	USA	GU227844	GU227942	GU228236	GU228040	[26]
*C. lilii*	CBS 109214	*Lilium*	Japan	GU227810	GU227908	GU228202	GU228006	[26]
*C. lineola*	CBS 125337	*Apiaceae*	Czech	GU227829	GU227927	GU228221	GU228025	[27]
	CBS 125339	*Apiaceae*	Czech	GU227830	GU227928	GU228222	GU228026	[27]
*C. liriopes*	CBS 119444	*Lirope muscari*	Mexico	GU227804	GU227902	GU228196	GU228000	[28]
	CBS 122747	*Liriope muscari*	Mexico	GU227805	GU227903	GU228197	GU228001	[26]
*C. phaseolorum*	CBS 157.36	*Phaseolus radiatus*	Japan	GU227896	GU227994	GU228288	GU228092	[26]
	CBS 158.36	*Vigna sinensis*	Japan	GU227897	GU227995	GU228289	GU228093	[26]
*C. rusci*	CBS 119206	*Ruscus*	Italy	GU227818	GU227916	GU228210	GU228014	[28]
*C. spaethianum*	CBS 167.49	*Hosta sieboldiana*	Germany	GU227807	GU227905	GU228199	GU228003	[27]
	CBS 100063	*Lilium*	South Korea	GU227808	GU227906	GU228200	GU228004	[27]
*C. spinaciae*	CBS 128.57	*Spinacia oleracea*	The Netherlands	GU227847	GU227945	GU228239	GU228043	[26]
	CBS 108.40	*Spinacia oleracea*	The Netherlands	GU227848	GU227946	GU228240	GU228044	[26]
*C. tofieldiae*	CBS 495.85	*Tofieldia calyculata*	Switzerland	GU227801	GU227899	GU228193	GU227997	[29]
	CBS 168.49	*Lupinus polyphyllus*	Germany	GU227802	GU227900	GU228194	GU227998	[27]
*C. trichellum*	CBS 118198	*Hedera*	Guatemala	GU227813	GU227911	GU228205	GU228009	[26]
	CBS 448.90	*Hedera helix*	Germany	GU227814	GU227912	GU228206	GU228010	[26]
*C. truncatum*	CBS 151.35	*Phaseolus lunatus*	USA	GU227862	GU227960	GU228254	GU228058	[26]
	CBS 119189	*Phaseolus lunatus*	USA	GU227863	GU227961	GU228255	GU228059	[26]
	CBS 710.70	*Phaseolus vulgaris*	Brazil	GU227864	GU227962	GU228256	GU228060	[26]
	YN1932501	*Camavalia brasiliensis*	China	**OP616009**	**OP649740**	**OP649744**	**OP649748**	**This study**
	YN1932502	*Camavalia brasiliensis*	China	**OP616010**	**OP649741**	**OP649745**	**OP649749**	**This study**
	YN1932503	*Camavalia brasiliensis*	China	**OP616011**	**OP649742**	**OP649746**	**OP649750**	**This study**
	YN1932504	*Camavalia brasiliensis*	China	**OP616012**	**OP649743**	**OP649747**	**OP649751**	**This study**
*C. lindemuthianum* (outgroup)	CBS 151.28	*Phaseolus vulgaris*	UK	GU227800	GU227898	GU228192	GU227996	[26]

## Data Availability

Not applicable.

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
