# Peer review of "Colletotrichum truncatum—A New Etiological Anthracnose Agent of Sword Bean (Canavalia gladiata) in Southwestern China"

_pathogens, 2022, doi:10.3390/pathogens11121463_

Round 1

Reviewer 1 Report

The manuscript describes initial work on the identification of anthracnose causative agent in sword beans as observed in the Yunnan province, China. The species causing leaf and stem spots was verified as C. truncatum by using sequence analysis of the ITS, ACT, GAPDH, and HIS3 genes. This was used to complement morphological and cultural characterization to identify this Colletotrichum species.

The manuscript is written well, with clear experimental logic and appropriate methodology applied to both pathogenicity assay and molecular characterization.

I have only one minor correction to suggest - in Line 105 the concentration of conidia is given as "1x106 conidia/ml"  which should be corrected to 1x106 conidia/ml"

Author Response

Dear editors and reviewers:

Thank you very much for your reasonable comments.

We have revised our manuscript based on your comments, details as follows:

1 There is some information that authors need to add, for example, alongside with ITS gene, why did authors choose ACT, gapdh, and HIS3 genes for multiple DNA sequencing?

Response: It was described that single gene ITS cannot clearly distinguish between C. lentis and C. truncatum (Xu, 2017), phylogenetic analyses based on the combination of ITS, ACT, HIS3, and GAPDH sequences clearly distinguished C. truncatum from the other closely related Colletotrichum species and described in the text.

2 The authors mentioned in lines 161 and 174 that “3) the fungus was subsequently re-isolated”, where is the evidence of the re-isolated strain, for example, colony morphology or DNA sequencing?

Response: The re-isolated strain from the inoculated plants identified as C. truncatum by colony morphology and was consistent with that previously described in Figure 2.

3 In Figures 1a, 1b and Figures 4, it will be easier if the authors add an arrow to indicate the symptom area.

Response: it is clearly added in revised text.

  1. The authors did not mention Figure 1c in the result and in Figure 2c, can authors add a scale bar?

Response: it has done.

5 Please change Figure 8 on line 149 to Figure 3 and change Figure 3 in lines 164 and 167 to Figure 4.

Response: it has done.

6 In Material and Methods, the authors mentioned using MEGA 7.0.2 (line 89), however, in the result authors wrote MEGA 5.1 (line 144), please clarify.

Response: it is clearly stated in revised text.

7 Table 2, please remove underlining in the table and add the header of the table on Page 6.

Response: it has done.

8 Please kindly check the scientific name throughout the Manuscript, it should be in Italic style, for example, lines 126, 151, 153, 163, 165, 167, 168, 216 and the spacing throughout the MS, lines, 31, 41, 46, 4, 56, 73 etc. Please check the font of 25oC in line 103, the format of references and affiliation.

Response: it has checked in revised text.

Thanks again.

                                      Shi Min, Li Yanzhong

                                        19- November-2022

Reviewer 2 Report

The authors explained the screening of Colletotrichum truncatum which was reported as a novel anthracnose agent of sword beans using morphological and multiple DNA sequences. However, there are a lot of mistakes that authors need to correct

1. In Figures 1a and 1b, it will be easier if the authors add an arrow to indicate the symptom area. 

2. The authors did not mention Figure 1c in the result. In my understanding, it should be on line 123, after ‘reach 100%’, isn’t it? 

3. Figure 2c, can authors add a scale bar? 

4. In Material and Methods, the authors mentioned using MEGA 7.0.2 (line 89), however, in the result authors wrote MEGA 5.1 (line 144), please clarify.

5. Please change Figure 8 on line 149 to Figure 3. 

6. Table 2, please remove underlining in the table and add the header of the table on Page 6. 

7. There is some information that authors need to add, for example, alongside with ITS gene, why did authors choose ACT, gapdh, and HIS3 genes for multiple DNA sequencing? 

8. Please change Figure 3 in lines 164 and 167 to Figure 4. 

9. In Figure 4, please add an arrow to indicate the symptom area. It is difficult to observe the lesion area.  

10. The authors mentioned in lines 161 and 174 that “3) the fungus was subsequently re-isolated”, where is the evidence of the re-isolated strain, for example, colony morphology or DNA sequencing? 

11. Please kindly check the scientific name throughout the Manuscript, it should be in Italic style, for example, lines 126, 151, 153, 163, 165, 167, 168, 216 

12. Please kindly check the spacing throughout the MS, lines, 31, 41, 46, 4, 56, 73 etc. 

13. Please check the font of 25oC in line 103. 

14. Please check the format of references. 

15. Please check Affiliation. 

Author Response

(The authors gave the same response as above.)

Round 2

Reviewer 2 Report

Please check the spacing of the word ex Line 201, 205. 

Author Response

Dear editors and reviewers:

Thank you very much for your reasonable comments.

We have revised our manuscript based on your comments, details as follows:

Please check the spacing of the word ex Line 201, 205.

Response: it has checked in revised text.

Thanks again.

                                      Shi Min, Li Yanzhong

                                        27- November-2022